# Structural Credit Assignment in Neural Networks using Reinforcement Learning

**Dhawal Gupta, Gabor Mihucz, Matthew K. Schlegel**
Department of Computing Science, Alberta Machine Intelligence Institute (Amii)
University of Alberta
{dhawal,mihucz,mkschleg}@ualberta.ca

**James E. Kostas, Philip S. Thomas**
College of Information and Computer Sciences
University of Massachusetts
{jekostas,pthomas}@cs.umass.edu

**Martha White**
Department of Computing Science
CIFAR AI Chair, Amii
University of Alberta
whitem@ualberta.ca

## Abstract

Structural credit assignment in neural networks is a long-standing problem, with a variety of alternatives to backpropagation proposed to allow for local training of nodes. One of the early strategies was to treat each node as an agent and use a reinforcement learning method called REINFORCE to update each node locally with only a global reward signal. In this work, we revisit this approach and investigate if we can leverage other reinforcement learning approaches to improve learning. We first formalize training a neural network as a finite-horizon reinforcement learning problem and discuss how this facilitates using ideas from reinforcement learning like off-policy learning. We show that the standard on-policy REINFORCE approach, even with a variety of variance reduction approaches, learns suboptimal solutions. We introduce an off-policy approach, to facilitate reasoning about the greedy action for other agents and help overcome stochasticity in other agents. We conclude by showing that these networks of agents can be more robust to correlated samples when learning online.

## 1 Introduction

Training neural networks involves *structural credit assignment*: attributing credit (or blame) to nodes in the network for correct (or incorrect) predictions. The output from a node early in the network impacts all the outputs downstream and finally the prediction outputted at the end of the network. Our goal is to adjust the weights that produced the output for this node, so that the prediction would have been more accurate. The most widely used solution for the structural credit assignment problem is backpropagation [44], namely gradient descent on the loss for the outputs.

Moving beyond backprop provides more flexibility in training neural networks. Backprop requires differentiability of activations and losses for the network, as well as synchronicity for computing the gradient and updating the weights. To update a node internal to the network, a full feedforward and backward pass needs to be computed, with global gradient information sweeping backwards from the output. Ideally, for online agents operating real-time, with computational constraints, we would have nodes that update each step, locally and asynchronously.

To make progress towards this lofty goal, we revisit an old idea: treating each node as an agent. Work in reinforcement learning (RL), including ideas like eligibility traces, were in fact inspired by Klopf [23] and the hedonistic neuron. It is not surprising that the idea of using an RL agent for each node is

found in early work, including the original REINFORCE algorithm [61], which is a policy gradient approach using sampled returns. Most work treating each node as an agent uses the REINFORCE update, often with baselines for variance reduction, including work on learning with spiking neurons [14] and CoAgent Networks (CoANs) [56; 55; 24]. The work in CoANs 1) nicely formalizes the idea of a collection of agents—each agent corresponding to a node or subset of nodes—cooperating to maximize return and 2) provides a general theorem on the validity of using the REINFORCE update. For this reason, we adopt their terminology and use CoANs to refer to networks composed of agents.

More recently, other algorithmic ideas from reinforcement learning, beyond REINFORCE, have begun to affect training of (stochastic) neural networks. The ideas of critics and baselines, which reduce the variance of policy gradient updates, have been well-developed for stochastic computation graphs [60]. This work provides a unification of gradient derivations, but as yet not an investigation into practical algorithms for structural credit assignment in neural networks. Other work on learning under stochastic neurons has typically used REINFORCE as a basic method, and explored other heuristics to improve learning, such as straight-through estimators [8], rather than improved RL approaches. Other work on credit assignment is loosely inspired by the idea of bootstrapping in RL, including synthetic gradients [20; 26] and fixed-point propagation [36].

Overall, however, the broader space of RL algorithms has not been leveraged to learn CoANs. One reason for this omission could be that the structural credit assignment problem within the neural network has not been clearly defined as an RL problem; rather, it was simply intuitive to use REINFORCE approaches for each node. Even the theory from the original CoANs work focused on the return in the environment—since CoANs were used to solve a reinforcement learning problem— and did not explicitly formalize the structural credit assignment problem within the network. Another reason could be that many straightforward ideas are not effective, as we show in this work.

To facilitate the use of RL algorithms, we first formalize the structural credit assignment problem as a finite horizon RL problem. We show local policy gradient updates provide an unbiased estimate of the joint gradient for structural credit assignment, ensuring REINFORCE is a sound approach. We then discuss key ideas from RL—namely exploration and off-policy learning—that can be leveraged to improve learning in CoANs. We show that REINFORCE can train multi-layered networks, but faces issues with suboptimality due to coagents learning under nondeterminism of fellow coagents. We provide an in-depth study highlighting this problem and measuring the entropy of different parts of the network. This in-depth study motivates the difficulties in using the common on-policy approaches, and we discuss and show how off-policy learning is a more promising direction. Finally, we discuss the advantages of CoANs when moving away from the standard iid learning setting, showing it can perform better than backprop on a continual learning problem with a highly correlated dataset.

## 1.1 Other Related Work

The literature on approaches to structural credit assignment is vast, with much of it using ideas different from reinforcement learning. One category of approaches uses local updates to make activations similar to a target vector of activations, such as target propagation [7; 28], the method of auxiliary variables [9] and fixed point propagation [36]. Kickback approximates the backprop update for ReLu networks, using an approximation to the gradient that allows for local updates [5]. Feedback alignment [31; 38] involves using random weights, instead of the actual weights in the next layer, that avoids symmetric propagation that is thought to be biologically implausible. Weight perturbation and node perturbation approaches have been used to estimate gradients, with node perturbations emerging as the preferred approach [46]. Fiete and Seung [14] showed that their node perturbation algorithm actually includes REINFORCE as a special case, linking these two classes of methods. However, the connection only exists for Gaussian noise perturbations and REINFORCE; the connection is lost for other perturbations as well as for other RL algorithms.

Other work has focused on changes in weights across time. Spiking neural networks and the associated spike-timing-dependent plasticity (STDP) learning rule [32] adjusts weights based on the relative timing of activations for nearby nodes. Equilibrium propagation [47] involves a phase of propagation in the network until reaching a low energy state, followed by a learning update. This work showed similarities to STDP, Contrastive Hebbian Learning [35] and Contrastive Divergence [18]. Reinforcement learning updates have been used for spiking neural networks, called Reward-modulated STDP. These updates use a global reward but with local update rules [29; 30], with some interesting insights that perturbations can be beneficial to induce exploration [30]. This area has

focused on delayed reward, namely assigning credit from node changes across time and across multiple updates. The local updates use node perturbation with eligibility traces to link perturbations on this step, to accuracy (rewards) at later time steps [30; 34]. A more recent algorithm, called cross propagation [58], explicitly adjusts weights back-in-time, to account for accuracy on this step, similarly to some meta-learning strategies but completely online.

There has also been some work using REINFORCE to learn activation paths through a network [13] and learning when to activate parts of the network [8; 6]. Other work has connected structural and temporal credit assignment, but in the opposite direction from this work: specifying temporal credit assignment as a structural credit assignment problem [4].

Once we use RL agents as nodes, which have stochastic policies, there is a clear connection to the work on stochastic neural networks. Much of this work has looked at networks with stochastic binary activations [37; 8; 42; 33], though the wider literature on stochastic computation graphs encompasses a broad range of stochastic neural networks [52; 48; 60]. Early work considered an EM-style algorithm [52] and a simple alternative, called the straight-through estimator [8], that directly passes the gradient back through the node to the weights that created the activation. The straight-through estimator has recently been shown to be a valid estimator for stochastic binary networks [49], and a lower-variance update has been proposed [17].

Multi-agent reinforcement learning approaches tackle a similar problem, in the cooperative setting. The coagents in the neural network can be seen as a group of agents cooperating to produce accurate predictions, though with the notable difference that there is an ordering to the actions taken by coagents. A common approach in this area has been to use reinforcement learning agents, and consider mechanisms for coordination without centralization. Some approaches have been to carefully define rewards for each agent [63; 62]; to use a single global reward plus some noise [11]; to use independent Q-learners [51]; or to estimate a global critic [39] potentially with local policy updates [15; 43]. When using independent Q-learners for each agent, the other agents are treated as part of the environment; consequently, the environment appears non-stationary. Hyper Q-learning [54] reduces the impact of this non-stationarity by estimating other agent's policies. These strategies do not directly extend to CoANs, but the connection to the cooperative multi-agent reinforcement learning problem could provide fruitful avenues for improved algorithms.

Finally, there have been several works empirically investigating alternative update strategies and architectures. Spiking neural networks generally have lower accuracy than standard deep neural networks, but a recent study has shown that with advances in hardware and algorithms to train spiking neural networks, this gap has become smaller [53]. Stochastic neural networks, particularly with binary activations, have been shown to be difficult to train, but with some improvements to the gradient estimator, can have significant advantages, including providing a level of regularization [42]. In this work, we aim to provide a more comprehensive empirical investigation into the use of reinforcement learning approaches to train CoANs.

## 2 Structural Credit Assignment as a Finite Horizon RL Problem

In this section, we describe how to formalize the credit assignment problem in a feedforward neural network as a finite-horizon RL problem. We start in the simplest setting, where we have a fully connected feedforward neural network composed of $k$ layers of size $n$. We assume we have inputs $x \in \mathcal{X}$, prediction targets $y \in \mathcal{Y}$, hidden layer activations $a_j = f(z_j)$ for pre-activations $z_j$ with activation function $f$. The basic idea for the finite horizon formulation is that the horizon is the number of layers, and on each step the input state for the agent is the activations from the previous step and the output is the activations for this layer or the final output prediction.

Consider the following agent-environment interaction. On the first step, given the sampled input $x \in \mathcal{X}$, the initial observation is $o_0 = [x, 0]$ where the 0 in the observation indicates the step-index in the finite horizon problem. The agent observes input $o_0$ and then takes action $a_0 \in \mathcal{A}$ that is a vector of activations (or pre-activations $z_0$)—these actions can be discrete or continuous. The next observation is deterministically $o_1 = [a_0, 1]$ (or $o_1 = [f(a_0), 1] = [f(z_0), 1]$). Then the agent inputs $o_1$ and outputs the activations for the next layer $a_1$ and obtains next observation $o_2 = [a_1, 2]$ (or $o_2 = [f(a_1), 2]$). This transition is Markov, because given $o_1$ and $a_1$, the next observation $o_2$ is independent of $o_0$. This interaction continues for $k$ steps, terminating at the last layer when the final action is to output the prediction $\hat{y}$. We depict this interaction in Figure 1.

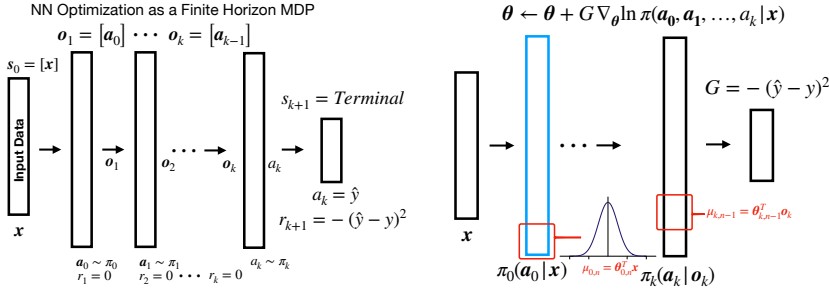

Figure 1: Structural Credit Assignment within a Neural Network as a Finite Horizon RL problem. The **left** figure maps components in the network to observations, actions and rewards. The **right** figure highlights one agent in the network and how it might be parameterized.

For this interaction to be Markov, the computation of the final reward requires the input $\boldsymbol{x}$. The rewards received during the episode are zero, with a reward given upon termination that is a function of the error between prediction and target, such as $\text{err}(\hat{y}, y) = (\hat{y} - y)^2$ with reward $-\text{err}(\hat{y}, y)$. The probability of $y$ depends on $\boldsymbol{x}$, and so the reward depends on $\boldsymbol{x}$ on this last step. We can obtain Markov states by simply including $\boldsymbol{x}$ in each observation. This part of the input can be ignored by the policy, as it is for most NN architectures. Formally, finite-horizon undiscounted MDP consists of state space $\mathcal{S} = \mathcal{X} \times \{0\} \cup \mathcal{X} \times \mathbb{R}^n \times \{1, \ldots, k-1\}$, actions $\mathcal{A} = \mathbb{R}^n$, where each start state $\boldsymbol{s}_0 = [\boldsymbol{x}, 0]$ and later states for $j \geq 1$ are $\boldsymbol{s}_j = [\boldsymbol{x}, \boldsymbol{a}_{j-1}, j]$. The transitions are deterministic, with a reward of zero on each step until termination at step $k$ giving reward $r_{k+1} = -\text{err}(\hat{y}, y)$.

A typical RL agent learns a separate policy for each horizon. This corresponds to learning one stationary policy, because the step-index is included in the state. Practically, however, these (stochastic) policies $\pi_j(\boldsymbol{a}_j | \boldsymbol{o}_j)$ for $j \in \{0, \ldots, k\}$ are updated separately, without considering the single stationary policy. If the agent is using policy gradients, then it uses parameterized distributions for the policies, such as Gaussians with means parameterized by the activations from the last layer. These policies can be updated using a REINFORCE update, using the gradient of the CoAN objective.

The CoAN objective corresponds to a policy gradient objective. An episode trajectory, with index information implicit and assuming actions are activations, is

$$\boldsymbol{o}_0 = \boldsymbol{x}, \boldsymbol{a}_0, r_1 = 0, \boldsymbol{o}_1 = [\boldsymbol{a}_0], \ldots, \boldsymbol{o}_k = [\boldsymbol{a}_{k-1}], a_k = \hat{y}, r_{k+1} = -\text{err}(\hat{y}, y), \text{Termination}.$$

Because $\pi(\boldsymbol{a}_0, \boldsymbol{a}_1, \ldots, \boldsymbol{a}_k | \boldsymbol{x}) = \pi(\boldsymbol{a}_0 | \boldsymbol{x}) \prod_{j=1}^{k} \pi_j(\boldsymbol{a}_j | \boldsymbol{a}_{j-1})$, the probability of this trajectory is $p(\boldsymbol{x}) \pi(\boldsymbol{a}_0, \boldsymbol{a}_1, \ldots, \boldsymbol{a}_k | \boldsymbol{x}) p(y | \boldsymbol{x})$. This is because $p(\boldsymbol{x})$ gives the probability of the start state; $\pi(\boldsymbol{a}_0, \boldsymbol{a}_1, \ldots, \boldsymbol{a}_k | \boldsymbol{x})$ gives the probability of the trajectory from $s_0$ because the state outcomes are deterministic given the action (activation or pre-activation); and $p(y | \boldsymbol{x})$ defines the probability of the reward on termination, since the target is stochastic. This provides a straightforward policy gradient objective, for undiscounted return $G \stackrel{\text{def}}{=} -\text{err}(a_k, y)$

$$\max_{\pi} \int p(\boldsymbol{x}) \pi(\boldsymbol{a}_0, \boldsymbol{a}_1, \ldots, \boldsymbol{a}_k | \boldsymbol{x}) p(y | \boldsymbol{x}) \, G \, d\boldsymbol{x} \, d\boldsymbol{a}_0 \, \ldots \, da_k \, dy = \max_{\pi} \mathbb{E}_{\pi, p(\boldsymbol{x}, y)}[G] \qquad (1)$$

If we assume each policy $\pi_j$ has parameters $\boldsymbol{\theta}_j$, then the stochastic gradient of this objective separates out into the following stochastic gradients for each policy separately: $G \nabla_{\boldsymbol{\theta}_j} \ln \pi_j(\boldsymbol{a}_j | \boldsymbol{o}_j)$. We show this formally in Proposition 1 in Appendix A. A similar result has been shown for CoANs used in the RL setting [24, Theorem 1] and for stochastic computation graphs [60, Theorem 2]; we include the result specifically for this case because it avoids much of the complications from those other works.

The locality of the policy gradient means policies can be updated locally with their own gradients, with a shared global return signal. We can also easily incorporate control variates to reduce the variance of the gradient, called baselines. The update is $(G - V(\boldsymbol{o}_j)) \nabla_{\boldsymbol{\theta}_j} \ln \pi_j(\boldsymbol{a}_j | \boldsymbol{o}_j)$, where the learned baseline $V(\boldsymbol{o}_j)$ estimates expected return given $\boldsymbol{o}_j$: $G - V(\boldsymbol{o}_j)$ corresponds to the advantage for the actions selected. Several different critics have been proposed for stochastic computation graphs [60]; we discuss how to make similar baselines for this finite horizon problem in Appendix B.

The CoAN is trained as a stochastic network, but is evaluated under the standard (deterministic) backprop setting. The question we ask is: can a CoAN, trained with local updates, produce predictions with similar accuracy to those produced by an NN, trained with backprop? We therefore test the

CoAN using the greedy actions from the co-agents, to provide deterministic predictions. Effectively, we are comparing the parameters $\boldsymbol{\theta}$ produced by the CoAN to those produced by backprop.[1]

**Remark:** We described the formalism for the simpler feedforward setting, to facilitate understanding. All the ideas, however, extend to more generic acyclic networks, because every acyclic network has a topological ordering on nodes. The state input for each coagent still consists of the input nodes. At each time step, whichever coagents have all their nodes evaluated—namely have their state input available—can produce their action. This propagates forward until a prediction is produced.

## 3 Issues with Stochasticity in On-policy Updates for CoANs

In this section, we show that though CoANs with REINFORCE can learn, they are significantly hindered by stochasticity in other coagents. Variance is a well-known problem in stochastic networks. Here, we highlight that even for a variety of variance reduction approaches, learning plateaus at a suboptimal point. The issue is less severe with discrete actions—binary rather than continuous activations—but a significant gap remains when contrasting to idealized (low-variance) gradients.

### 3.1 Experimental Details

We investigate CoANs on problems where backprop is known to perform well, to provide a strong baseline and facilitate understanding the behavior, and potential issues, when learning in CoANs. We expect backprop to outperform CoANs here, and ask: how much worse are CoANs compared to backprop, and can we close the gap with simple variance reduction techniques? To investigate this question we use two well-studied datasets: MNIST [27] for classifying handwritten digits, and the Boston Housing Dataset from UCI.

Where possible, we matched the architecture and optimization choices for backprop and the CoAN learner. We test both strategies using a single and double-layer neural network, with 64 hidden nodes and ReLU activations. Each node in the CoAN is a single coagent using a Gaussian distribution with parameterized mean and a fixed standard deviation, set system-wide through a systematic sweep over $\sigma \in \{0.1, 0.5, 1.0, 2.0, 4.0, 8.0, 16.0\}$. The feedforward procedure samples actions from these policies and then applies the layer's activation function (i.e. ReLU for hidden layers or the identity/softmax function for the output layer). Both use RMSProp [57], with fixed $\beta = 0.99$ and stepsizes swept for $\alpha \in \{2^{-7}, 2^{-9}, 2^{-11} \ldots, 2^{-15}\}$. We use mini-batch gradient descent with batch size 32 for MNIST with 50 epochs, and full gradient descent for the Boston Housing Dataset with 10k epochs. Hyperparameters are chosen from performance on a validation set held out from the training set: 10K for MNIST and 51 samples for Boston Housing. Results are averaged over 10 independent runs and compared using the area under curve (AUC).

Along with the standard CoAN with no baseline, we also test the effectiveness of three baselines. The **Global baseline** (Co-G) maintains a scalar running average of the global loss function parameterized by $\eta$ the rate of decay (fixed at 0.99). The **State Global baseline** (Co-SG) learns a parameterized value function associated with each input / state to the complete network and tries to predict the loss. The **State Layer baseline** (Co-SL) is a per layer baseline, where we learn a parameterized value function for each layer based on its input learned using Monte Carlo updates from the global loss. More details on baselines are in Appendix B and pseudocode in Appendix H.

### 3.2 Results

We report the performance of the best performing parameters on a held-out test set with the same size as the validation set in Figure 2 (a). At a glance, it is clear that backprop outperformed all the CoANs. It is surprising that there is no improvement using the local baselines as compared to the global baseline. A natural question from these results is if the gap between CoANs and backprop is due

---

[1] The optimal parameters for the (surrogate) policy gradient objective above may not be the same as those for the deterministic network. Namely, the co-agents minimize $\mathbb{E}[(\hat{Y} - y)^2]$ across pairs $(\boldsymbol{x}, y)$, where $\hat{Y}$ is stochastic due to the stochastic co-agent policies. This is in contrast to minimizing the deterministic output $(\mathbb{E}[\hat{Y}] - y)^2$ or the loss where each co-agent takes a greedy action. The co-agents may actually adjust their outputs, to mitigate risk from the stochasticity in fellow co-agents. Somewhat surprisingly, we find in our experiments that there is no such gap, and optimizing this surrogate can produce optimal solutions.

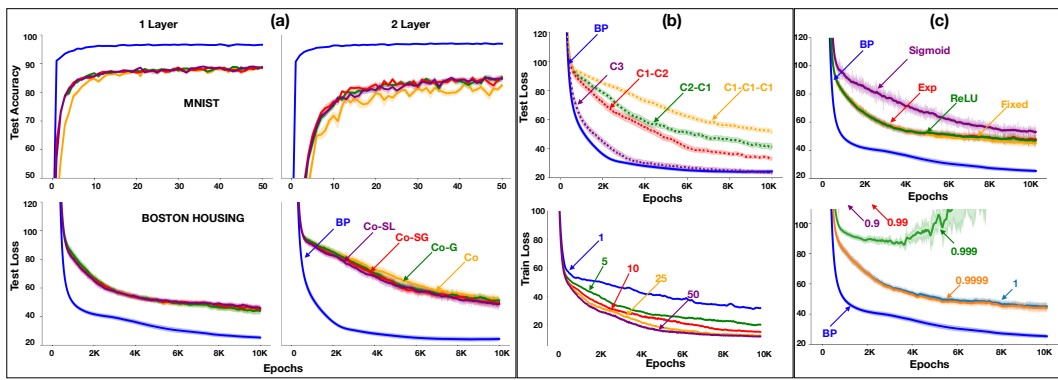

Figure 2: All curves are averaged for 10 runs with standard error bars. **(a)** Training different variants of CoAN's and backprop on MNIST and Boston Housing with one and two layer NNs. **(b)** Experiments highlighting issues with nondeterminism in coagents. The upper graph depicts performance with different coagent partitionings of the network. The lower graph depicts the training curve for 2 layer coagent network with C1-C1-C1 partitioning for different amounts of averaging i.e., $[1, 5, 10, 25, 50]$. **(c)** Failures of different strategies to gradually reduce nondeterminism of coagents overtime, on Boston Housing.

to the optimization process, or a poor stationary point of the CoAN itself. Experiments running the CoAN for longer showed only very slow improvement. Before this experiment, we hypothesized that high variance updates would negatively impact the CoAN optimization, but the variance reduction schemes used above did not improve performance at all.

To investigate the role of stochasticity in coagents, we test different partitioning schemes of the two-layer coagent network. We can treat the whole network as one coagent, and use backprop within that agent (called C1). We can use two coagents: one that outputs the activations for the final layer and one that learns the weights to produce the prediction (called C2-C1). We have four partitioning schemes, where we always have a prediction coagent to keep the objective consistent, labeled: C1-C1-C1, C2-C1, C1-C2, and C3. C1-C1-C1 has a coagent for each layer, and C1-C2 uses a linear coagent for the first layer and a single layer neural net coagent for the next.

In Figure 2 (b, upper), we see the deterministic network C3 performs similarly to the backprop network. As we add stochasticity back to intermediary layers, performance degrades significantly. This highlights that the stochasticity in the last layer is not a culprit, and that variance is lower for C1-C2 where stochasticity is in the first layer than for C2-C1 where stochasticity is in a later layer. In either case, the introduction of stochasticity in intermediate layers starkly reduces performance.

To ascertain that the issue is due to variance, rather than poor stationary points, we can use an idealized gradient that is not practical for our desired learning setting, but can act as a control for this experiment. We can obtain a low-variance gradient estimate simply by sampling the stochastic network many times, for the given pair $(\boldsymbol{x}, y)$. Such an update is non-local—requires significant coordination—and so is not a practical approach to reduce variance for our setting. But this averaged gradient, particularly with increasing samples for the average, gives insight into the optimization surface for these networks as well as the magnitude of the variance. Figure 2 (b, lower) shows the effect on learning with increasing samples used in the average gradient, for the two layer network. With increasing samples, the training of the CoAN significantly improves, nearly matching the learning speed of backprop with 50 samples. This highlights that the surrogate objective used by the CoAN results in reasonable solutions when testing the greedy actions given by the CoAN. It shows that there is significant variance in the gradient, and that baselines are not reducing that variance.

### 3.3   Failures of Simple On-policy Approaches to Reduce Nondeterminism

A natural next step is to consider simple strategies to gradually reduce the stochasticity of coagents. For example, it is common in RL to have a decay schedule for the exploration parameter. Similarly here, instead of using a fixed variance parameter for coagents, we can gradually decay this parameter and so allow coagents to gradually learn good actions under nearly greedy actions from other coagents.

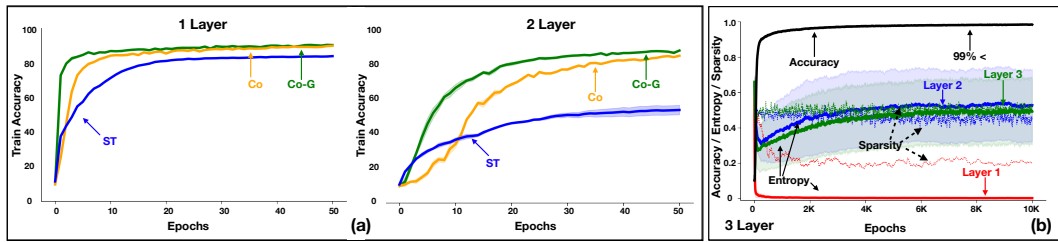

Figure 3: Experiments on MNIST, using 64 nodes per layer. **(a)** Learning curves for discrete agents (ST, Co, Co-G) with single and double-layers, averaged for 10 runs, for 50 epochs. **(b)** Learning progress of three layer network for 10K epochs, with entropy *(solid line)* and sparsity *(dotted line)* of each layer, along with accuracy (scaled to $[0, 1]$), which matches backprop (with accuracy $> 99\%$).

We tested two strategies to enable the variance parameter in the Gaussian distributed actions to decrease with time. The first strategy is to learn a $\sigma$ per node, using a bias unit per node, passed through either a sigmoid, ReLU or exponential to obtain a non-negative number. The second is to use a decay rate for the $\sigma$ of the whole network. In both cases we initialized $\sigma$, with the best value found in the original experiments (i.e. $\sigma = 4.0$). Unfortunately, these natural strategies to remove stochasticity in coagents do not improve performance, as can be seen in Figure 2 (c). One possible reason is that for smaller $\sigma$ of $0.9$ and $0.99$ the gradient $\nabla_{\boldsymbol{\theta}} \ln \pi(\boldsymbol{a}|\boldsymbol{o})$ can get very big for low probability sampled actions. This instability can be avoided with discrete actions, or with off-policy learning, both of which we investigate next.

### 3.4 Discrete Actions Helps Control Stochasticity, But Not Enough

Here we study discrete coagents with Bernoulli discrete actions, and their stochasticity over time. Discrete networks provide an easier way to handle stochasticity in their policy as a softmax parameterization adapts stochasticity when learning, and can become fully deterministic. As a baseline, we do gradient backpropagation via straight-through (ST) estimators [8; 50], because standard backprop cannot be used for discrete nodes. In this set of experiments, we again use REINFORCE coagents, and test on MNIST. We measure the entropy of the coagents over time, as well as the sparsity of the representation to ensure reasonable levels of activation.

In Figure 3(a) we can see that CoANs actually perform better than the baseline, the ST estimator, and here the baseline had a bigger positive effect (Co-G versus Co). ST estimators have difficulties when learning in more than a single layer [8], probably due to misalignment [50]. For Co-G, the ability to better control the entropy seems to help: in Figure 3(b) the drops in entropy—earlier for layer one and then at 200 epochs for layer two and three—cause a sudden rise in accuracy. However, the entropy does not fully decrease and there is some unintuitive behavior. The first layer becomes deterministic faster, which is surprising as it relies on stochastic actions of downstream agents. The entropy for layer two and layer three also decreases, but eventually, the entropy starts to creep back up while maintaining or slightly improving the accuracy. We note that these networks are able to match the performance of a standard NN with backprop (i.e., accuracy $> 99\%$) but it takes a very large number of training steps (approximately $10K$ epochs).

We provide a more in-depth investigation into the stochasticity under discrete actions, including using more discrete actions per coagent and training with (randomized) subsets of coagents fixed, in Appendix C. Overall, we find that the former significantly increases performance in 1-layer networks for both REINFORCE coagents and especially action-value methods; however, the latter approach does not substantially reduce the entropy, nor improve performance.

## 4 Off-Policy Learning to Learn Nearly Deterministic CoAgents

The chief difference between using RL and typical optimization approaches, both for SCGs and standard NNs, is that we can learn off-policy. When training a neural network, it is rarely the case that the accuracy of predictions matters when doing an update. Rather, this setting matches the fully offline learning setting—the pure exploration setting—instead of the online setting where the agent needs to maximize reward while learning. This highlights that we can also use many different

exploration approaches, to gather useful data about how to adjust the policies for more accurate predictions. The behavior could choose to make a poor prediction, to gather experience that is more useful for improving accuracy than if the on-policy best prediction was used. This separation is in stark contrast to methods like backprop.

In this section, we show how to learn critics off-policy, and that this improves on using on-policy critics. We start by describing the algorithm, and then provide results in MNIST for the case of discrete coagents. We provide additional results in Appendix F showing that on-policy action-value methods do not improve on REINFORCE, further motivating the shift to off-policy learning.

## 4.1 An Off-Policy Algorithm for CoANs

The first step is to modify the REINFORCE update, to use an action-value critic. Instead of using a sampled return $G$, we can estimate the expected return $Q_j(o_j, a_j)$, for the coagent taking action $a_j$ given input $o_j$, namely the previous hidden layer $a_{j-1}$. The update then uses $(Q_j(o_j, a_j) - V_j(o_j))\nabla_{\theta_j} \ln \pi_j(a_j|o_j)$, where for the final layer we do not learn a critic and simply using the immediate reward (i.e., the negative of the error). These action-values can be updated on-policy, each time the network is queried, using a Sarsa update

$$\theta^{(q)}_j \leftarrow \theta^{(q)}_j + \alpha(0 + Q_{j+1}(o_{j+1}, a_{j+1}) - Q_j(o_j, a_j))\nabla Q_j(o_j, a_j)$$

where $Q_j(o_j, a_j)$ can simply be a linear function of $o_j, a_j$—namely a linear function of $[a_{j-1}, a_j]$—or could itself be a small neural network.

This strategy, however, introduces bias for two reasons. First, estimating action-values means we have some error in our expected return estimate, due both to estimate error and approximation. Second, the local action-values are actually tracking a non-stationary target. They estimate the expected return for an action, where the expectation is taken over the input and output as well as the actions of the other coagents. Further, the coagent is attempting to learn how to select actions, given stochastic action selection by the other coagents rather than the best action (greedy action) for each co-agent. This nonstationarity and difficulties in credit assignment is well-recognized as an issue in multi-agent reinforcement learning [54; 15; 43]. However, in our setting the known structure between agents means we can more easily obtain a solution, than an unstructured collection of cooperating agents.

The key is to reason about greedy actions of downstream coagents, rather than the action they actually took. The update has a small modification, to instead use a maximum over values in the next layer

$$\theta^{(q)}_j \leftarrow \theta^{(q)}_j + \alpha(0 + \max_{a'} Q_{j+1}(o_{j+1}, a') - Q_j(o_j, a_j))\nabla Q_j(o_j, a_j)$$

Given the input, the agent asks: what is the value of each action, given the maximal actions will be taken for downstream layers? This update bootstraps only on the action-value in the next layer, but the update for that $Q_{j+1}$ also bootstraps off of the max in the next layer. Therefore, each action-value starting from the end of the network is learning about maximal action-values for downstream coagents, and propagating that information backwards. This approach directly exploits the known Markov structure of the credit assignment problem, and so should learn more efficiently than using structureless algorithms like REINFORCE.

The coagents do not need to reason about the greedy actions for upstream coagents, because action-values are *conditioned* on inputs produced by those coagents. For a given activation from the previous layer $a_{j-1}$, the coagent learns $\pi(a_j|a_{j-1})$. Under sufficiently high capacity policy parameterizations, the coagent can simply learn what to output for a variety of different inputs, including those that are more optimal for the given input $x$. It is straightforward to show that in Equation (1), under unrestricted policy parameterizations, each policy can be optimized assuming these greedy action-values for downstream layers.

Practically, however, our coagent policies are likely to be under-parameterized. If an upstream coagent provides a wide range of activations $a_{j-1}$, then the policy has to trade-off function approximation across this large input space. Ideally, it would focus function approximation on the $a_{j-1}$ that are likely to produce good predictions $\hat{y}$. This trade-off is a common issue in policy gradient methods. We can incorporate a reweighting, to increase the importance of an update for an $a_{j-1}$ that is more probable—more greedy—from the upstream coagent, and decrease the importance of an update for lower probability $a_{j-1}$. Such importance is already implicit, because $a_{j-1}$ are sampled proportionally to the probabilities. For this reason, we do not address it further here, but note that there are some insights for state reweightings in policy gradient methods to improve solutions [21; 19].

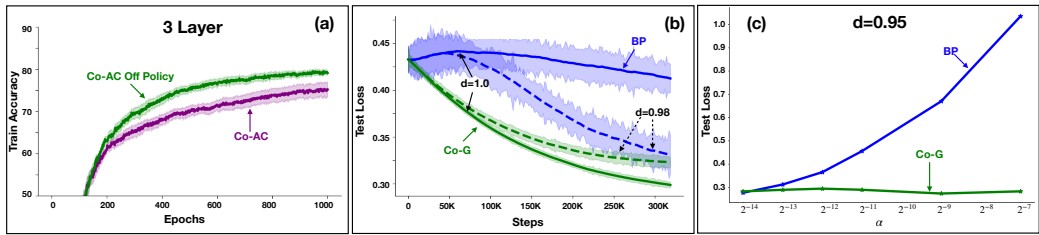

Figure 4: **(a)** Learning curves comparing on policy and off policy actor critic. Averaged for 10 runs, for 3 layer neural network with 64 nodes per layer. **(b)** CL experiments with problem difficulties at $d = 0.98$ *dashed* lines and *solid* lines for $d = 1.0$ comparing backprop and CoAN's. **(c)** Sensitivity plots in CL problem for learning rate ($\alpha$), for problem difficulty of $d = 0.95$.

## 4.2 Results with Off-Policy CoANs

We perform a set of experiments where we estimate $Q_j(\boldsymbol{o}_j, \boldsymbol{a}_j)$ function for each layer in the CoAN, using the on-policy and off-policy approaches described in Section 4.1. We allowed our agents to select from different forms of critics which include linear and single-layered neural nets with (64, 128 & 1024) nodes. We found that NN critics worked best with 1024 nodes and a learning rate twice the value of the CoAN. We keep the other hyperparameters the same from Section 3.4. Note that here our goal is to understand if critics can help, and so allow for these larger critics per coagent; practically, depending on the setting, there may be stronger limitations on the critics.

Figure 4 (a) presents the results for on-policy (**Co-AC**) and off-policy (**Co-AC Off Policy**) critics. We find that off-policy critics help improve performance, particularly later in learning, whereas the on-policy critic is plateauing at a lower point. As a note, the performance with critics is actually worse than that of REINFORCE, likely due to bias in the critics. Better approaches to learning the critic off-policy, including using methods like replay, should help us close this gap. Our goal in this experiment was primarily to contrast the on-policy and off-policy critics, to highlight that off-policy learning is a promising direction towards addressing the nondeterminism issue in CoANs.

## 5 Continual Learning Experiments

In the previous sections, we chose settings where backprop is effective, to make the results more interpretable and start from the standard learning setting. The motivation for CoANs, however, goes much beyond the standard setting. The goal is to facilitate learning in other settings, particularly in continual learning settings with correlated data and the need for real-time learning and decision-making. CoANs also naturally facilitate asynchronous inputs and updating [24], as well as learning with recurrence. Traditionally stochastic methods have known to regularize and help avoid local minima; we provide a small experiment showing this advantage of CoANs in Appendix D.

In this section, we investigate the utility of CoANs for prediction when learning online with highly temporally correlated inputs. We expect CoANs to be less prone to failure than backprop, which can completely overwrite previous learning when learning on correlated data. Because coagents are RL agents, the policies should better track and adapt to changes in the environment.

To measure coagents' ability to learn online with various degrees of temporal correlation, we examine the performance of Co-G on the PieceWise Random Walk Problem introduced in [41], with the same parameter settings and target function. In this dataset, the correlation difficulty parameter d ∈ [0,1] controls the amount of temporal correlation within the data: 0 stands for iid data points, while 1 indicates a fully temporally correlated dataset.

We first train Co-G and backprop for 180,000 training steps, with d ∈ $\{0, 0.85, 0.95, 1\}$, and test it every $900^{th}$ step on a test set of 1,800 iid samples, with the best hyperparameters picked on the validation set of equal size. Standard deviation for Co-G is swept $\sigma \in \{2^{-4}, 2^{-2}, 2^0, 2^1, 2^2, 2^3, 2^4\}$. For both algorithms, stepsize values are swept $\alpha \in \{2^{-14}, 2^{-13}, 2^{-12}, 2^{-11}2^{-9}, 2^{-7}\}$, the batch size is fixed at 32, number of units at 50, and number of hidden layers at 1. Co-G is trained using the RMSProp optimizer, while backprop is trained using the Adam optimizer, both with standard $\beta$ values. Then, using the chosen hyperparameters, we allow the agent to continue learning up to 320,000 training steps on d ∈ $\{0.98, 1\}$, to gauge the long run performance on extremely correlated data. Results are averaged over 200 runs.

Backprop outperforms Co-G on iid samples, and achieves lower error on d=0.85. Co-G begins to display an edge at d=0.95, where it achieves similar performance to backprop, and at d=1, Co-G performs better and learns steadily, while backprop is incapable of learning. In addition, the sensitivity plot for this experiment in Figure 4 (c) depicts that while backprop's performance is largely dependent on the correct alpha parameter, Co-G's performance remains approximately the same across the swept alpha values. Co-G is also similarly insensitive to its $\sigma$ hyperparameter, as shown in the Appendix E.

Looking at long run performance after 180,000 steps, we see two interesting phenomena. Co-G is able to continue to steadily learn on the highly correlated dataset, unlike backprop. But, for d=0.98, backprop actually starts to match the performance of Co-G. These results, however, are for carefully swept hyperparameters, and we see that backprop is quite sensitive to the stepsize. Under this heavy correlation, therefore, we find that CoANs provide more robust performance, in that they provide steady progress from the very beginning of learning and are much less sensitive to the stepsize.

## 6    Limitations and Next Steps

Once we have formalized this problem as a finite-horizon RL problem, it facilitates applying a variety of RL algorithms. The approaches in this work are arguably the simplest first choices, and do not incorporate useful advances in exploration strategies, policy gradient methods, off-policy learning and replay approaches. Our CoANs relied solely on the stochasticity in the coagents to explore. Such randomized exploration strategies are unlikely to be as efficient as directed exploration approaches, even when restricted to only those that learn values and not models, such as those using upper confidence bounds on values [16; 40; 25] and information-directing sampling [45]. Such directed exploration strategies have also recently been developed for policy gradient methods [2].

There have also been many recent advances to policy gradient methods, with better theoretical understanding and improvements for the off-policy setting. Of particular relevance is work that examines the importance of state weightings and dealing with distribution shift [3], which is key for the off-policy setting [19]. Another important insight is leveraging connections between approximate policy iteration and policy gradient methods, to obtain performance guarantees [12; 1; 59; 10]. This perspective assumes that an explicit policy and explicit values are learned, and facilitates off-policy learning using the learned values. One of these methods, called SBEED [12], further exploits recent advances in gradient-based methods for learning values off-policy.

All of these more advanced approaches rely on value estimation. Value estimates help direct exploration, by facilitating reasoning about uncertainty and incorporating optimism. They naturally facilitate off-policy learning and replay, because they allow us to learn from short experience tuples. They allow each agent to reason about counterfactual outcomes, and perform more policy updates in the background, asynchronously. The bias, however, currently precludes obtaining these benefits; we as yet do not know why it is so harmful. When using Actor-Critic in other settings, the bias in the critic does not seem to be so detrimental. One issue here may be that PG methods have not been specially designed for finite-horizon problems, where a different policy is used each step. For us, the changing policies make it less straightforward to use standard bias-reduction strategies to estimate critics, like eligibility traces. Understanding the influence of bias in the critic for structural credit assignment, and how to overcome it, is one of the most important open questions for CoANs.

## 7    Conclusion

In this paper, we investigated the use of reinforcement learning (RL) for the structural credit assignment problem in neural networks. We formalized this problem as a finite-horizon MDP, and showed that local policy gradient updates for each node (coagent) provide an unbiased estimate of the joint policy gradient for all nodes. We show that the basic local policy gradient update for this coagent network (CoAN) can learn—even under difficult learning settings like highly correlated data—but that it plateaus at suboptimal solutions. Through a set of targeted experiments, we highlight that the stochasticity amongst the coagents results in this suboptimality, and that it cannot be mitigated with standard variance reduction strategies or attempts to gradually reduce stochasticity of the coagents towards deterministic policies. We highlight that off-policy learning can naturally be applied to this problem, through the use of off-policy critics. We show that this strategy is promising, but that much more work needs to be done to improve the learned critics and mitigate bias.

## Acknowledgments and Disclosure of Funding

This work was supported by NSERC Discovery, IVADO, CIFAR through CCAI Chair funding and by the Alberta Machine Intelligence Institute (Amii). This research was funded in part by NSF award #2018372. The authors would also like to thank Kirby Banman.

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
