# OpenReview forum: "Structural Credit Assignment in Neural Networks using Reinforcement Learning"
_NeurIPS.cc/2021/Conference — NeurIPS 2021 Poster_

### Official Review · Reviewer_JMW8 · 2021-06-29

**Rating:** 5
**Confidence:** 4

**Summary:**

This paper studies credit assignment problem in neural networks with RL algorithms. They first formulate the problem as a finite-horizon MDP so that they can apply RL. They find nondeterminism in the problem causes suboptimal solutions and propose to use off-policy RL with a critic to resolve it.

**Limitations And Societal Impact:**

It seems that the authors only test on very simple neural nets. My question would be: is this the tradition in this area or it is a proof of concept? If the latter, I believe more experiments with modern architecture is encouraged.

All the figures are not self contained. For example, some of the legends are missing. The difference between dashed lines and solid lines are not clearly readable within the figures.

Setting, experimental results and analysis are intertwined together without clear separation.

The results are not showing the potential for utilizing the proposed technique in other areas.





**Main Review:**

This paper's main contribution is to understand off-policy RL is better than the previous policy gradient method (REINFORCE) in credit assignment problems. The authors did a thorough analysis and explained well about the previous difficulty and why they introduce critic into this problem. In the view of an RL researcher, the algorithmic contribution is quite limited. Hence, the significance of this paper is not high enough for the bar of this conference.

But from the angle of credit assignment, these might be crucial findings and opens a welcoming avenue for future research. My judgment might not be correct due to the fact I only know RL well.

As the authors mentioned, there might be multiple aspects from RL that can be applied to this problem as well. My suggestion would be to add those to the current framework as well and provide somewhat more convincing results would make this paper stronger.

**Time Spent Reviewing:**

2

---

> ### Author Response · Authors · 2021-08-09
> **Response to Reviewer JMW8**
>
> Thank you for reading our paper and providing useful comments.
>
> We indeed did not provide any groundbreaking algorithmic contributions to the field of RL in this paper, however as you alluded to in your review as well: this paper's main contribution is to investigate structural credit assignment (SCA) using RL approaches. The novelty is in identifying how to leverage and improve RL algorithms for a collection of RL agents for SCA, and understanding the limitations of what seem like straightforward approaches. This work (a) highlights that different variance reduction approaches are insufficient, (b) identifies the issue of nondeterminism, from a series of targeted experiments and (c) proposes a strategy to use off-policy learning to address that issue, with some supporting evidence for the promise of that approach. We additionally demonstrate that CoANs can actually outperform backprop on a temporally correlated dataset.
>
> A next step, which we are currently pursuing, is to investigate additional RL techniques to  improve our off-policy approach. As mentioned to reviewer brEu, we will include a more thorough discussion on possible directions, including more generally how to leverage recent advances in RL such as those for policy-gradient methods and in exploration. This paper lays a foundation for further exploration into SCA as an RL problem, and it warrants a broader discussion on how RL algorithms and advances in RL could be  leveraged.
>
> **It seems that the authors only test on very simple neural nets. My question would be: is this the tradition in this area or it is a proof of concept?**
>
> For scientific experiments, it is typical to start in simpler settings. We chose simpler neural networks, because they were sufficient for these problems and backprop is able to learn high accuracy solutions. This can be seen as a proof of concept, rather than a tradition, though many papers on stochastic neural networks also have their experiments on simple neural networks.
>
> One of our current next steps is looking at CoANs and backprop for much deeper neural networks. Such experiments, naturally, are more time consuming. This first investigation, where we ran many different algorithms and configurations, warranted smaller problems and networks to facilitate a careful investigation.
>
> **All the figures are not self contained. For example, some of the legends are missing. The difference between dashed lines and solid lines are not clearly readable within the figures.**
>
> We will work to make the figures clearer in the final version of the paper.
>
> **Setting, experimental results and analysis are intertwined together without clear separation.**
>
> We treat the experimental analysis and results as a way to guide the reader to the choices made throughout the experimental process. We start with the most straightforward investigations and analyze the results followed by steps that can be taken to explore hypotheses built on the initial results.
>
> As mentioned to another reviewer, we chose to do “just-in-time” explanations of experiments and algorithms, precisely because there are quite a few details that can be hard to remember across sections. We tested quite a few algorithms (baselines, partitioning, pretraining, critic-based approaches), with different architectures (discrete and continuous) and different problem settings. We originally put all algorithms, including the off-policy ones, up front and found it was more onerous for the reader.

---

> > ### Comment · Reviewer_JMW8 · 2021-08-19
> > **Rebuttal Response**
> >
> > After reading the rebuttal, I don't think the authors resolve my core concern. Hence, I will keep my score.

---

### Official Review · Reviewer_wqi1 · 2021-07-12

**Rating:** 6
**Confidence:** 3

**Summary:**

This paper presents a study that investigate RL methods for the credit assignment problem in neural network. In this study, the credit assignment problem is formalized as a finite horizon RL problem. Subsequently, it is empirically demonstrated that stochasticity of coagents’ policies hinders the learning performance when REINFORCE is used to train coagents. To address this issue, a Q-learning like off-policy algorithm is proposed. The experimental results show that the proposed off-policy algorithm outperformed on-policy algorithm. In addition, the proposed off-policy algorithm outperformed backpropagation in a continual learning task.

**Limitations And Societal Impact:**

Although this study presents a method for credit assignment in neural networks, the results do not indicate that the backpropagation will be replaced with the CoAN in near future. I recommend to clearly describe this point for audience who are not experts in this field.

**Main Review:**

Strong points:
-	The empirical results on the effect of stochasticity are interesting.
-	The proposed algorithm outperformed backpropagation in the continual learning task.

Weak points:
-	The paper is not well-organized.
-	Some more experimental results are necessary to support the claims.

I think that the empirical results on the stochasticity of coagents are interesting, but the paper is not well-organized and hard to follow in some parts. I think that the paper requires major revision to improve the clarity. I recommend to separate the algorithmic discussion and the experimental section, instead of writing finding in a chronological order.
In addition, it is necessary to add some more experiments to support claims.  I listed points to be improved in the following.

Detailed comments
-	Kostas et al. [2020] also formalized the credit assignment problem as RL, and I do not see clear difference between the formulations in this paper and their work.
-	Presentation in this paper is not well-organized, and it is hard to follow in some parts. For example, I’m not sure whether the action is assumed to be discrete or not in Section 4. If the action is discrete, it is natural to apply the Q-learning algorithm without an explicit model of a policy. Please clarify whether the action is discrete or not and discuss the relation to the Q-learning algorithm and “Co-AC off-policy”, which is proposed in Section 4.1.
-	In Figure 2, I guess MC represents on-policy Monte Carlo, but there is no description that explains what MC represents.
-	In equations below the line 304 and 317, I do not understand we have
 … \alpha( 0 + max Q … instead of \alpha ( r + max Q…
-	In the continual learning experiment, the proposed method is referred to as “Co-G”. I expected that “Co-AC off-policy” should be evaluated in this experiment. In addition, variants of RL methods, including “Co-AC on-policy” and “on-policy Monte Carlo”,  should also be evaluated in the continual learning task.

=== comments after author response ===
As my questions were answered in the author response and discussions, I raised the score. Although I'm leaning to acceptance after discussions, I would like to remind the author that the paper requires major revision to improve the clarity.



**Time Spent Reviewing:**

3

---

> ### Author Response · Authors · 2021-08-09
> **Response to Reviewer wqi1**
>
> **Kostas et al. [2020] also formalized the credit assignment problem as RL, and I do not see clear difference between the formulations in this paper and their work.**
>
> Our focus is structural credit assignment (SCA), and formalizing the credit assignment problem internally as an RL problem. The focus from Kostas et al. was on using CoANs to solve RL problems (both structural and temporal credit assignment), and they did not explicitly formalize the SCA problem as a finite horizon problem. They still obtain a similar policy gradient result, where gradients can be computed locally at each node, because that naturally occurs for the product of probabilities over activations at each node. However, by explicitly formalizing the SCA problem as a finite horizon RL problem, we can then leverage RL algorithms to improve learning internally in the network. This was not done by Kostas et al.
>
> **Presentation in this paper is not well-organized, and it is hard to follow in some parts. For example, I’m not sure whether the action is assumed to be discrete or not in Section 4. If the action is discrete, it is natural to apply the Q- learning algorithm without an explicit model of a policy. Please clarify whether the action is discrete or not and discuss the relation to the Q-learning algorithm and “Co-AC off-policy”, which is proposed in Section 4.1.**
>
> The RL formalism allows for discrete or continuous action sets. We begin by allowing the action set to be generic (either discrete or continuous). Then, for a specific architecture chosen for the NN, the action set is concretely specified. For ReLU activations, we used continuous actions. For binary activations, we used discrete actions. RL algorithms have been developed for both discrete and continuous actions, therefore depending on the NN specification, different algorithms are more appropriate.
>
> You are correct to point out that using the Q-learning algorithm essentially requires that the actions be discrete, so as to efficiently calculate the max operator (though there are a few algorithms for continuous action Q-learning). In this work, when applying Q-learning, we restrict to the discrete action architectures; we will clarify this in the paper.
>
> As for the connection to Co-AC, the Co-AC on-policy algorithm uses an on-policy Sarsa update for the critic and Co-AC off-policy uses the off-policy Q-learning update for the critic. Note that, for a node, both updates actually update the on-policy action taken by that node, but reason differently about other nodes. The on-policy critic refers to the fact that we evaluate the on-policy (executed) actions by the other nodes, and the off-policy critic refers to the fact that we evaluate assuming the greedy action is taken by other nodes.
>
> **In Figure 2, I guess MC represents on-policy Monte Carlo, but there is no description that explains what MC represents.**
>
> Monte Carlo (MC) was added as a baseline, to include a comparison to a value-based method that gets to use full returns (like REINFORCE). You are right that we did not clearly explain this goal, nor the terminology. We had provided the algorithm for MC in the Appendix; we will refer to it in the main paper and will explain the reason for this addition.
>
> **In equations below the line 304 and 317, I do not understand we have … \alpha( 0 + max Q … instead of \alpha ( r + max Q…**
>
> Our intermediate rewards, internal to the network, are r_{j+1} = 0; we simply explicitly replace the r with a 0 to show this. Only the last update, on termination, has a non-zero reward.
>
> **In the continual learning experiment, the proposed method is referred to as “Co-G”. I expected that “Co-AC off-policy” should be evaluated in this experiment. In addition, variants of RL methods, including “Co-AC on-policy” and “on-policy Monte Carlo”, should also be evaluated in the continual learning task.**
>
> The purpose of this experiment was to compare CoANs and backprop in this different problem setting. For that reason, we chose one of the basic CoAN algorithms for the comparison. A reasonable next question is to ask how the different CoAN algorithms compare on this problem. Here, we did not want to detract from our primary question comparing CoANs and backprop, and the behavior with the simplest algorithm already highlights the improved robustness to correlation that can be obtained with CoANs.
>
> **Although this study presents a method for credit assignment in neural networks, the results do not indicate that the backpropagation will be replaced with the CoAN in near future. I recommend to clearly describe this point for audience who are not experts in this field.**
>
> This is a good idea. In the extra page available for camera ready, we can include a longer discussion about current limitations, that CoANs are as yet not a viable alternative for standard training methodologies in standard supervised learning setups and what might be needed to change that. This interfaces well also with the suggestion from another reviewer to discuss what algorithms could be leveraged from RL.
>
> As a final comment about presentation, we will incorporate the clarifications you highlighted and consider how to make algorithms and experiments accessible. In several cases, confusion may have arisen simply from omitted details; we will carefully sweep through the paper to identify any such areas. We chose to do “just-in-time” explanations of experiments and algorithms, precisely because there are quite a few details that can be hard to remember across sections. We tested a decent number of algorithms (baselines, partitioning, pretraining, critic-based approaches), with different architectures (discrete and continuous) and different problem settings. We originally put all algorithms, including the off-policy ones, up front and found it was more onerous for the reader. We believe we can maintain the separation we currently have, and still address your concerns on organization and clarity.

---

> > ### Comment · Reviewer_wqi1 · 2021-08-18
> > **Thank you for the clarification**
> >
> > Thank you for the response. My questions are clarified in the author response. However, concerns regarding the presentation still remain because it is hard to assess how much the presentation will be improved after the revision.

---

### Official Review · Reviewer_To4V · 2021-07-14

**Rating:** 6
**Confidence:** 3

**Summary:**

The paper investigates using reinforcement learning to train neural networks. Specifically, it considers formulates the problem in which individual layers are treated as steps in a finite horizon partially observable MDP, and explores using either a global agent or individual co-operating agents at each layer to maximize the reward (the negative final error).

The paper then presents an empirical study of using REINFORCE in this cooperative agent framework to train one or two hidden layer networks on MNIST/Boston Housing while varying different parameters like the policy variance, different partitioning schemes of the network into agents, different baseline approaches to improve stability, and activations. The key takeaways of this experiment are that (1) the CoAN+REINFORCE approach generally underperforms backprop, (2) baselines to improve stability don't seem to help, and (3) the primary reason for the poor performance is the non-deterministic behavior of the other agents (as indicated by having a single agent optimizing all layers performs the best by far).

Motivated by this, the paper presents an off-policy Q learning approach to training cooperative agents, which uses bootstrapping and trains by assuming the best action is taken at the next step. While the paper shows this improves over on-policy SARSA, it still underperforms REINFORCE, which underperforms backprop.

Lastly, the paper shows that in a continual learning setting the CoAN+REINFORCE approach can be more robust than backprop.

**Limitations And Societal Impact:**

No discussion of broader/societal impact, though not sure how much is needed for this work.

**Main Review:**

*Strengths*
- The paper studies a fascinating topic in using RL to train networks, and the abstract/introduction does a great job in motivating the potential for local/asynchronous training and the possibility of using it for computationally efficient and more general training.
- While I am less familiar with the prior work in this area, the related work effectively summarized prior approaches and seemed thorough.
- The empirical study shown in this work on CoANs+REINFORCE is interesting, and sheds some light on what makes the problem challenging (even if the results are still worse than backprop)
- Similarly, the formulation of using off-policy Q learning in the CoAN framework to address the issue of is a creative and to the best of my knowledge novel way of approaching the problem.
- The experiment in the continual learning setting is interesting and suggests that in the case of highly correlated data points using the CoAN framework may be more robust.

*Weaknesses*

(1) The clarity and presentation of the empirical study in Section 3 could be improved. There is a lot going on, and the takeaways are currently disorganized. The main takeaway seems to be that the stochastic behavior of different agents is the main source of poor performance from the plot comparing C1-C1-C1 vs C3, but its difficult to follow within all of the other experiments.
more specifically:
- Figure 1 is difficult to follow, particularly with one grid (despite (b) and (c) being completely different experiments). I'd suggest splitting this up if possible and then having more informative captions/legends which explain what the plot is showing.
- The explanation of the "baselines" was very rushed, with just a single sentence each. Without more explanation it is not clear what exactly to take away from Figure 1(a) besides that CoAN+REINFORCE is worse than backprop. Also the baselines seem to be using extra additional parameters? Overall it was not clear what the baselines were meant to show/what the takeaways are.
- I also found the takeaways from Section 3.4 to be unclear - it makes sense that the straight-through optimizer performs poorly, but does the discretization of nodes actually help CoAN+REINFORCE? From the plots in Fig 1(a) vs Fig 2 (a) they look pretty similar.

(2) I found the Q Learning for CoANs to be an interesting and well motivated approach. However it does seem that the critics need an additional network during training which is optimized separately which seems to be a strong limitation - essentially requiring multiple other networks to train the original network. Even with this extra model capacity it also appears to underperform the initial REINFORCE approach, so its not clear how promising this approach is.

(3) While the continual learning experiment setup is interesting, the CoAN+REINFORCE only appears better in the most extreme cases of correlation [0.98, 1], and its not clear if these results hold generally.

**Time Spent Reviewing:**

2.5

---

> ### Author Response · Authors · 2021-08-09
> **Response to Reviewer To4V**
>
> Thank you for your helpful comments! We address each concern below.
>
> **(1) The clarity and presentation of t Section 3: Figure 1**
>
> Figure 1 summarizes all the experiments for continuous action coagents. We felt this appropriately gives the reader a full view of the results, and is space efficient. However, since you highlight that it is difficult to follow, we will absolutely rethink how it is presented, and improve the labeling and captions for these plots.
>
> **The explanation of the "baselines"**
>
> We define and explain the baseline in detail in the appendix. As stated in section 3.2, we initially believed reducing update variance would improve the network’s performance. The actual results refute this hypothesis. These results are informative for understanding where we should develop improvements for CoANs, but the baseline algorithms were not of importance for the rest of the paper. The goal of these experiments was to show that REINFORCE with even the more complex baselines, that reduce variance more than the basic global baseline, insufficiently improved performance.
>
> **Section 3.4...does the discretization of nodes actually help CoAN+REINFORCE**
>
> In short, yes. With discrete nodes, coagents achieve an accuracy of 99% within 10k epochs. With continuous activations, coagents were not able to obtain that level of accuracy in 10k epochs; note that in the paper we report performance for a shorter training time for continuous activations, but we also experimented with 10K epochs and we observed continuous to not attain the performance attained by discrete nodes
>
> **(2) I found the Q Learning for CoANs to be an interesting and well motivated approach. However it does seem that the critics need an additional network during training which is optimized separately which seems to be a strong limitation - essentially requiring multiple other networks to train the original network. Even with this extra model capacity it also appears to underperform the initial REINFORCE approach, so its not clear how promising this approach is.**
>
> While we do show critics can improve performance of CoANs in the off-policy setting, the reviewer is correct to point out that the current approach is limited. The goal of these experiments was to identify if the use of off-policy critics could improve performance, and so direct future research towards improving how we learn those critics. Conceptually, the idea is sound, and it is clear we need to move beyond the basic REINFORCE update and exploit other RL approaches to train CoANs; such off-policy approaches are one relatively broad step in that direction. There are many avenues for improvements to learning these critics, and as mentioned to reviewer brEu, we will include a more thorough discussion on possible such directions.
>
> **(3) While the continual learning experiment setup is interesting, the CoAN+REINFORCE only appears better in the most extreme cases of correlation [0.98, 1], and its not clear if these results hold generally.**
>
> Backprop is able to leverage a lot of structural information, for structural credit assignment. We generally expect backprop to perform better, when it performs reasonably. For this benchmark, it seems that backprop was effective under relatively high levels of correlation. What is interesting here is that when backprop begins to fail, the CoAN continues to be able to learn.

---

> > ### Comment · Reviewer_To4V · 2021-08-22
> > **Re: Response**
> >
> > Thanks for the response:
> >
> > Re (1):
> > I understand presenting all the results in a space efficient way is a challenge, but would recommend some iteration on the presentation of the plots. Also the result of the discrete agents actually outperforming continuous after 10K steps is valuable, I would recommend explicitly showing this result more clearly in the paper.
> >
> > Re (2):
> > It is an interesting result that the off-policy critics show some benefits over on-policy CoANs, suggesting that leveraging off-policy data can be valuable for this problem. I still have some concerns about the scalability of this approach, since it requires another Q network with additional weights to be trained effectively, which is itself can be a challenging optimization problem. But perhaps there is potential for the secondary network to be smaller in parameter count, in which case the method could lead to performance.
> >
> > Re (3): Got it, this makes sense.
> >
> > Overall, I think the results of the paper are interesting, even if the proposed methods generally still underperform backprop. Assuming that the authors are able to address some of the presentation/clarity issues (especially Figures 1 and 2b), and can add some more of the above discussion of limitations of the Q learning approach, I'm raising my score to a 6.

---

### Official Review · Reviewer_brEu · 2021-07-15

**Rating:** 7
**Confidence:** 4

**Summary:**

- This paper revisits an interesting (and truly under-explored) idea from the reinforcement-learning literature: coagent networks [43,44]. The original work by Philip Thomas considered endowing the standard agent, that maps from states to actions, with a richer structure whereby the state can be mapped by multiple coagents both in parallel and in sequence yielding various intermediate outputs (coagent actions) before ultimately rendering a final action to constitute the action output of the overall agent.

- In this paper, the authors consider how the concept of coagent networks may be used to ground neural network training as a finite-horizon reinforcement-learning problem, thereby offering a local, asynchronous alternative to the traditional and widespread method of backpropagation.

- The paper offers a formulation of neural network training as a finite horizon POMDP, where the partial observability arises only to accommodate the non-Markovian nature of the terminal reward signal that must depend on the initial input to the network. While the formulation allows the authors to employ policy-gradient techniques for local, asynchoronous optimization of the network, they highlight shortcomings of using the standard REINFORCE estimator, which align with early findings by [44].

- On the path to resolving these issues, the authors present numerous experiments that tease apart various hypotheses to diagnose the failures of coagent network training relative to standard backprop. While unable to outperform backprop in the standard supervised learning setting, the authors not only provide ample diagnosis of the failure but also conclude their empirical investigation with a continual learning setup that highlights the strength of using local, adaptive policies for neural network training over backprop.

**Limitations And Societal Impact:**

The authors are quite upfront in their discussion of the limitations in the proposed coagent network approach and adequately highlight paths forward for future work to consider in bypassing these obstacles.

**Main Review:**

This paper scores very highly on significance, quality, and clarity. I commend the authors for the writing style and structure of the paper which gives the reader a sense of the authors' actual thought process when designing each experiment in light of observed results and falsified hypotheses (for instance, L225-233); this is often something valuable that goes unwritten in many research papers. On the axis of clarity, I'm not especially familiar with the literature on alternatives to backpropagation, however the authors do seem to offer a detailed account of prior work in Section 1.1.

I think the meticulous empirical investigation in this paper is something to be excited about as it opens up numerous possibilities for bringing recent advances in RL theory and practice to bear on the problem of efficient, adaptive neural network training. Even though the authors have not succeeded in providing a coagent network setup that outperforms backprop in the standard supervised learning setting, the continual learning experiments are compelling and this paper on the whole will initiate conversations and spark interest from the RL community to help address the outstanding performance gap.

As a minor point, I did notice that the authors typically refer to the problem formulation as a finite-horizon MDP throughout the text whereas the actual derivation itself specifies the need for a POMDP setup (L144-152). I wonder if perhaps this is unnecessary and the authors might simply consider a non-stationary state space and transition function so that, in the H-1 stage, the state space can be augmented to include the input and all transitions moving from step H-1 to H must fold the input back into the state, leaving it available for the standard Markov reward computation. Another possibility might be to treat the input data as a goal and model training over an entire dataset as a multi-task RL problem [5].

I think there are various pieces of the reinforcement-learning literature that may be useful to think about in future work that builds off the problem formulation presented in this paper such as recent advances in provably-efficient policy gradient methods [1,2], information-directed exploration techniques based on Bayesian reinforcement learning and posterior-sampling methods [3,4], and much more. That said, the authors seem well-positioned (if not, best-positioned) to set the stage for these kinds of discussions; a section in the appendix that outlines some of the enticing but loftier possibilities could be really thought-provoking and interesting.

===== Post-rebuttal =====

While I'm still excited by this paper, the comments from the other reviewers certainly suggest that the presentation and clarity could be improved. I have looked at work on conjugate MDPs and policy co-agent networks in the past and so, maybe, the trajectory of the paper unfolded a bit better for me when reading. Nevertheless, for this weakness in clarity alone, I'll reduce my score down to a 7 with the hopes that the authors incorporate suggestions from the other reviewers to boost the presentation for a broader RL audience in a camera-ready version.

Given the novelty of the problem formulation and the wide range of possibilities that future work might explore, I'm far less concerned by the significance issues raised by the other reviewers (performance relative to backprop and more parameters overall for the critic) and continue to advocate for accepting the paper.

[1] Agarwal, Alekh, Sham M. Kakade, Jason D. Lee, and Gaurav Mahajan. "On the theory of policy gradient methods: Optimality, approximation, and distribution shift." Journal of Machine Learning Research 22, no. 98 (2021): 1-76.

[2] Agarwal, Alekh, Mikael Henaff, Sham Kakade, and Wen Sun. "Pc-pg: Policy cover directed exploration for provable policy gradient learning." arXiv preprint arXiv:2007.08459 (2020).

[3] Osband, Ian, Benjamin Van Roy, Daniel J. Russo, and Zheng Wen. "Deep Exploration via Randomized Value Functions." J. Mach. Learn. Res. 20, no. 124 (2019): 1-62.

[4] Russo, Daniel, and Benjamin Van Roy. "Learning to optimize via information-directed sampling." Advances in Neural Information Processing Systems 27 (2014): 1583-1591.

[5] Schaul, Tom, Daniel Horgan, Karol Gregor, and David Silver. "Universal value function approximators." In International conference on machine learning, pp. 1312-1320. PMLR, 2015.


**Time Spent Reviewing:**

3

---

> ### Author Response · Authors · 2021-08-09
> **Response to Reviewer brEu**
>
> Thank you for your detailed summary and kind comments; we greatly appreciate your feedback.
>
> **As a minor point, I did notice that the authors typically refer to the problem formulation as a finite-horizon MDP throughout the text whereas the actual derivation itself specifies the need for a POMDP setup (L144-152). I wonder if perhaps this is unnecessary and the authors might simply consider a non-stationary state space and transition function so that, in the H-1 stage, the state space can be augmented to include the input and all transitions moving from step H-1 to H must fold the input back into the state, leaving it available for the standard Markov reward computation. Another possibility might be to treat the input data as a goal and model training over an entire dataset as a multi-task RL problem [5].**
>
> Your suggestion about formalizing this problem as a multi-task RL problem is a great suggestion; we will add this as an alternative direction in the discussion section. We specified the problem as an MDP, rather than a POMDP, to start, to make it more accessible. But, it does make the actual specification less precise, when we then switch between them. We will instead do as you suggest: augment the state and discuss the problem as a finite-horizon MDP. Then, we will highlight that in most NN architectures the input data is not used for most nodes, meaning that the agents themselves introduce a small amount of partial observability.
>
> **I think there are various pieces of the reinforcement-learning literature that may be useful to think about in future work that builds off the problem formulation presented in this paper such as recent advances in provably-efficient policy gradient methods [1,2], information-directed exploration techniques based on Bayesian reinforcement learning and posterior-sampling methods [3,4], and much more. That said, the authors seem well-positioned (if not, best-positioned) to set the stage for these kinds of discussions; a section in the appendix that outlines some of the enticing but loftier possibilities could be really thought-provoking and interesting.**
>
> Excellent suggestion! We will include just such a discussion in the additional page allotted for the camera ready.

---

### Decision · Program_Chairs · 2021-09-27

**Decision:**

Accept (Poster)

**Comment:**

This paper started out with a strong accept and 3 weak rejects, but then converged to 7-6-6-5 (where reviewer with 5 has a short review and is less engaged than other 3 reviewers). I recommend the weak acceptance for the paper, primarily because the novelty/insights of the paper outweigh the immaturity of presentation/lack of pragmatic result (to outperform backprop). The formulation could be impactful for further research, and this paper lays out sufficient foundations for it. However, all reviewers agree that the presentation clarity must be substantially improved, and therefore the authors must follow through and incorporate the reviewer suggestions for the final version.